# Adult Trees *Cryptomeria japonica* (Thunb. ex L.f.) D. Don Micropropagation: Factors Involved in the Success of the Process

Alejandra Rojas-Vargas [1,2], Itziar A. Montalbán [2,†] and Paloma Moncaleán [2,*,†]

1 Instituto de Investigación y Servicios Forestales, Universidad Nacional, Heredia 86-3000, Costa Rica
2 NEIKER-BRTA, Department of Forestry Sciences, 01192 Arkaute, Spain
* Correspondence: pmoncalean@neiker.eus
† These authors contributed equally to this work.

**Abstract:** *Cryptomeria japonica* (Thunb. ex L.f.) D. Don is a commercial tree native to Japan and is one of the most important forest species in that country and the Azores (Portugal). Because of the quality of *C. japonica* timber, several genetic improvement programs have been performed. Recently, some studies focusing on *C. japonica* somatic embryogenesis have been carried out. However, in this species, this process uses immature seeds as initial explants, and for this reason, it is not possible to achieve the maximum genetic gain (100% genetic of the donor plant). Although some studies have been made applying organogenesis to this species, the success of the process in adult trees is low. For this reason, our main goal was to optimize the micropropagation method by using trees older than 30 years as a source of plant material. In this sense, in a first experiment, we studied the effect of different types of initial explants and three basal culture media on shoot induction; then, two sucrose concentrations and two light treatments (LEDs versus fluorescent lights) were tested for the improvement of rooting. In a second experiment, the effects of different plant growth regulators (6-benzylaminopurine, meta-topolin, and thidiazuron) on shoot induction and the subsequent phases of the organogenesis process were analyzed. The cultures produced the highest number of shoots when QL medium (Quoirin and Lepoivre, 1977) and long basal explants (>1.5 cm) were used; the shoots obtained produced a higher number of roots when they were grown under red LED lights. Moreover, root induction was significantly higher in shoots previously induced with meta-topolin.

**Keywords:** carbohydrates; conifers; cytokinins; LEDs; plant growth regulators; rooting; shoot induction





## 1. Introduction

The Japanese cedar, *Cryptomeria japonica* (Thunb. ex L.f.) D. Don, subfamily Taxodioideae, family Cupressaceae, is a monoecious conifer distributed across East Asia [1,2]. *C. japonica* covers approximately 4.5 million ha, representing 44% of the total reforested area [3]. This conifer is one of the most important timber tree species in Japan, and it is traditionally used for construction wood and for obtaining biomass [4].

Tsubomura and Taniguchi (2008) [5] mentioned that Japanese cedar is clonally propagated by cuttings, but this type of propagation requires many hours of manual labor, therefore it is difficult to establish a short-term propagation protocol. For the abovementioned reasons, biotechnological approaches for *C. japonica* clonal propagation, such as in vitro methods including somatic embryogenesis (SE) or shoot organogenesis, are valuable tools for the propagation of this conifer [6]. In this sense, SE is a recognized technique for the large-scale propagation of conifers [7]. However, SE is a complex, multistage process initiated from immature seeds, so it is not possible to reproduce the genotype of the donor plant. In contrast, micropropagation by nodal tissue culture is faster, and higher multiplication rates are possible [5,8]. Micropropagation starting from nodal segments uses smaller

explants than conventional techniques, and selection of juvenile explants is possible [8]. With regard to explants coming from mature trees, they are less likely to dedifferentiate and reprogram [9], but the use of explants from mature trees for conifer micropropagation has also been reported [10,11]. However, because these trees are selected after reaching maturity, there has been limited success reported in the vegetative propagation of mature conifers, and procedures should be improved.

Although the benefits of tissue culture for the propagation of forest trees have been recognized, the success of such methods is still highly dependent on the species, the explant quality, the age of the donor plant, the culture medium, plant growth regulators, and/or the interaction among all these different factors. As a result, morphogenesis determines the growth and development of plant tissues, and it is influenced by several physico-chemical factors [12]. In relation to culture media, DCR [13], MS [14], and QL [15] are commonly used basal media for in vitro regeneration of conifers [16,17]. Furthermore, the cytokinins added to the culture medium have a direct effect on the endogenous phytohormone balance, provoking a response to the induction of axillary shoot buds and affecting the organogenesis of the culture [6,18].

Carbohydrates in plants are basic elements; they constitute substrates for respiration and are essential for many other processes related to plant development or gene expression, and in many species they favor rooting, acting mainly as a source of energy [19–21].

Plant growth and development are also influenced by different physical factors, with light being one of the most important [22]. The traditional light source used in in vitro culture in the growth chambers is fluorescent tubes (FL), with irradiances between 25 and 150 mmol m$^{-2}$s$^{-1}$ for a 16 h photoperiod [23]. FL emit a broad light spectrum, and their physiological effects on plants are not specific [24]. Furthermore, the power consumption of FL is high as the heat emitted needs to be removed from growth chambers using air conditioners, making the process expensive [25]. Light-emitting diodes (LEDs) are available today as an alternative to conventional light sources for in vitro plant growth [26], and they present advantages over FL such as small size, longer lifespan, less power consumption, high energy conversion efficiency, and adjustable light spectra [27].

The analysis of the combined effects morphological (type of explants), chemical (culture media, sucrose concentrations), and physical factors (LED lights) at different micropropagation stages of adult trees has not been carried out in *Cryptomeria japonica*. Moreover, the use of ventilated culture containers is not widely used for conifer species. For these reasons, with the main objective of improving the micropropagation protocol for adult Japanese cedar, we focused on optimizing (1) the shoot induction stage using different types of explants, cytokinins, and culture media as well as (2) the rooting stage using different sucrose concentrations, Ecobox containers®, and light treatments (fluorescent versus LEDs).

## 2. Materials and Methods

### 2.1. Plant Material

Actively growing *C. japonica* twigs were collected in October 2019 from two healthy >30-year-old adult trees located in Arkaute (Spain; 42°51′9.35″ N, 2°37′30.55″ W) to carry out Experiment 1 (Figure 1a). In October 2020, actively growing twigs were collected from four healthy >30-year-old adult trees located in Urnieta (Spain; 43°13′13.03″ N, 1°57′40.91″ W), and one adult tree (Spain; 43°13′50.32″ N, 1°59′25.37″ W) was chosen to carry out Experiment 2.

### 2.2. Sterilization

The plant material was first washed with commercial detergent, then rinsed under running water for 5 min immersed in 70% ethanol for 2 min, and then washed two times with sterile distilled water in the laminar flow unit. Finally, the actively growing twigs were disinfected in commercial bleach (30% *v/v*) (active chlorine 37 gL$^{-1}$ sodium hypochlorite) for 20 min and rinsed three times in sterile distilled water for 5 min. each.

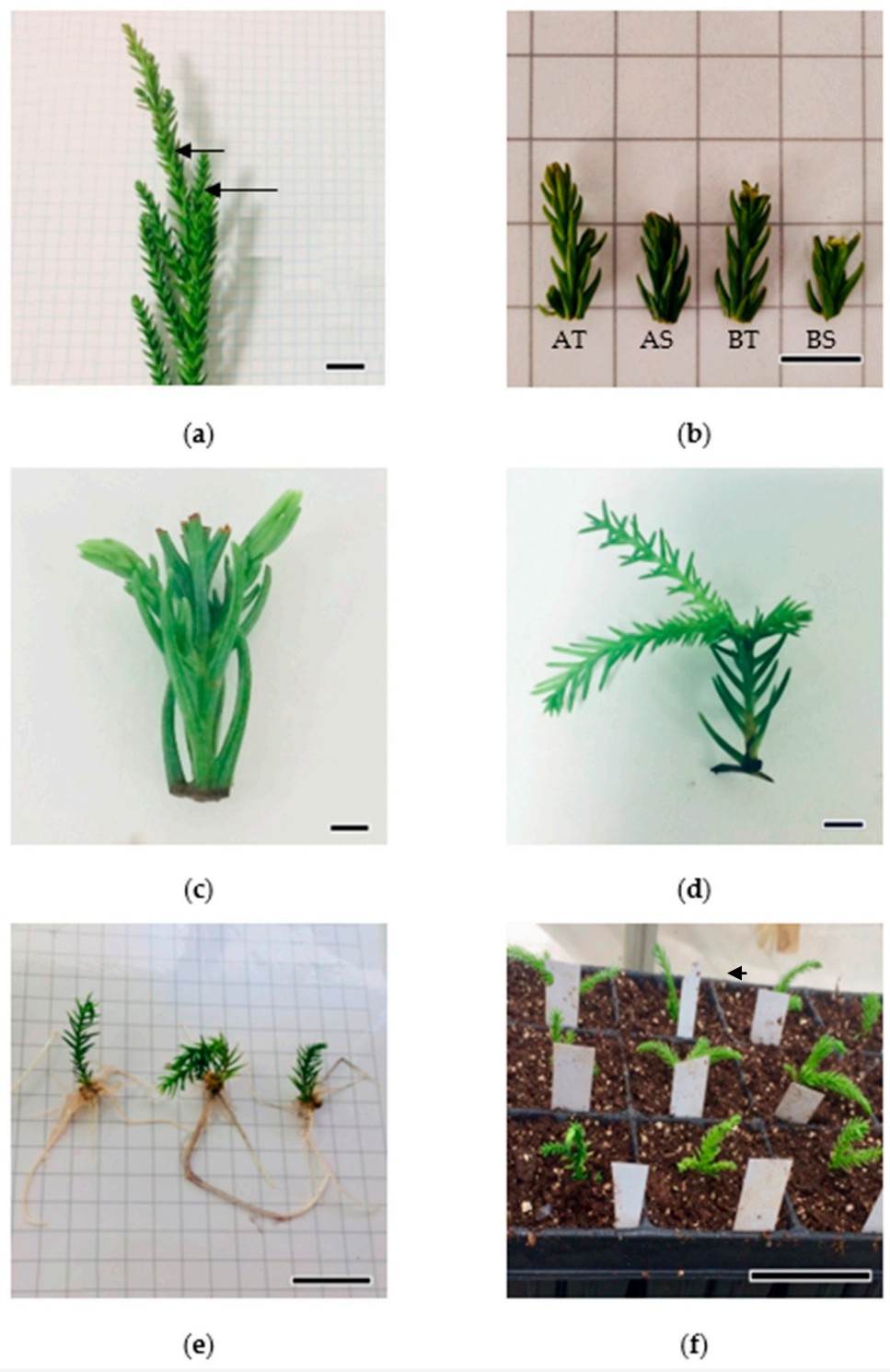

**Figure 1.** Plant material at different stages of *Cryptomeria japonica* micropropagation process: (**a**) actively growing twigs used as starting explants; arrows show the explants used for in vitro culture establishment, bar = 30 mm; (**b**) explant type; apical explants > 1.5 cm (AT), apical explants < 1.0 cm (AS), basal explants > 1.5 cm (BT) and basal explants < 1.0 cm (BS), bar = 10 mm; (**c**) basal explants of 2.0 cm length after 4 weeks cultured in QL medium [15] supplemented with 8.8 μM BA, bar = 30 mm; (**d**) elongated shoots after 6 weeks in hormone-free QL medium supplemented with 2 gL$^{-1}$ activated charcoal, bar = 30 mm; (**e**) rooted shoots after six weeks in hormone-free QL medium supplemented with 2 gL$^{-1}$ activated charcoal and under red LEDs, bar = 30 mm; (**f**) plantlets after four weeks in ex vitro conditions in the greenhouse, bar = 30 mm.

After the sterilization protocol, four types of explants were tested: apical explants > 1.5 cm (AT), apical explants < 1.0 cm (AS), basal explants > 1.5 cm (BT), and basal explants < 0 cm (BS) (Figure 1b).

*2.3. Micropropagation Process*

2.3.1. Experiment 1

After sterilization, the four types of explants were cultured vertically on $25 \times 145$ mm test tubes with polypropylene caps (Lab Associates, Oudenbosch, The Netherlands) containing 15 mL of bud induction medium (IM) (Supplementary Table S1). Three basal media were assayed: DCR [13], MS [14], and QL [15]. All media were supplemented with 3% (*w/v*) sucrose, 6-benzyladenine (BA, 8.8 µM, Duchefa Biochemie, Haarlem, The Netherlands), and 8 gL$^{-1}$ Difco Agar$^{®}$ (Becton and Dickinson, Madrid, España) granulated. The pH of all media was adjusted to 5.8, and then they were autoclaved at 121 °C for 20 min. All cultures were placed in the growth chamber at a photoperiod of 16 h with 120 µmol m$^{-2}$ s$^{-1}$ light intensity provided by cool white fluorescent tubes (TLD 58 W/33; Philips, Suresnes, France) and a temperature of $21 \pm 1$ °C.

As soon as shoot induction was observed (after four weeks) (Figure 1c), four to five explants were transferred to baby food glass jars with Magenta$^{TM}$ b-cap lids filled with 25 mL of elongation medium (EM) (Supplementary Table S1). EM was composed of hormone-free DCR, MS, or QL supplemented with 2 gL$^{-1}$ activated charcoal, 3% (*w/v*) sucrose, and solidified with 8.5 gL$^{-1}$ Difco Agar$^{®}$ granulated; pH and autoclaving conditions were those mentioned for IM.

The shoots were transferred to fresh EM every six weeks. Shoots were cultivated individually in a fresh EM when they reached 10–15 mm (Figure 1d). The conditions in the growth chamber were the same as those described above.

2.3.2. Root Induction and Acclimatization of Rooted Plants

Elongated shoots of at least 20–25 mm long were used for root induction. Based on the results of Experiment 1, QL basal medium was selected. The explants were transferred to Ecoboxes (Eco2box/green filter, consisting of a polypropylene oval vessel with a "breathing" hermetic cover, Duchefa$^{®}$) with 100 mL of root induction medium (RIM) (Supplementary Table S1), which was composed of half-strength macronutrient QL medium with 50 µM 1-naphthaleneacetic acid (NAA, Duchefa Biochemie, Haarlem, The Netherlands), 8 gL$^{-1}$ Difco Agar$^{®}$, and 3% sucrose or 1.5% (*w/v*) sucrose. The pH and autoclaving conditions were those previously described. The shoots were placed under dim light for eight days. Then, two different light treatments were tested for four weeks: (A) white fluorescent light (FL) (color temperature 4000 K), 120 µmol m$^{-2}$s$^{-1}$ light intensity provided by cool white fluorescent tubes (TLD 58 W/33; Philips, Suresnes, France); and (B) red light (peak wavelength 630 nm), 60 µmol m$^{-2}$s$^{-1}$ light intensity provided by adjustable LEDs (RB4K Grow Light LEDs). The photoperiod and the temperature of the growth chamber were the same as previously described.

After five weeks of culture in RIM, shoots were cultured for six weeks in Ecoboxes with 100 mL of root expression medium (REM) (Supplementary Table S1); this medium consisted of half-strength macronutrient QL medium supplemented with 2 gL$^{-1}$ activated charcoal, 3% sucrose or 1.5% (*w/v*) sucrose, and 8.5 gL$^{-1}$ Difco Agar$^{®}$. Then, the rooted plants (Figure 1e) were planted ex vitro, transferring them to moist peat moss (Pindstrup, Aarhus, Denmark) with vermiculite at a proportion of 8:2 (*v/v*); acclimatization was carried out in a greenhouse under controlled conditions at a temperature of $21 \pm 1$ °C and progressively decreasing the humidity during a month from 95 to 80% (Figure 1f).

2.3.3. Experiment 2

Based on the results of Experiment 1, basal explants of >1.5 cm length and QL medium were selected to perform this experiment (Supplementary Table S2). The medium was supplemented with one of these three types of cytokinin (CK): BA, meta-topolin (m-T), or

thidiazuron (TDZ, Duchefa Biochemie, Haarlem, The Netherlands), all of them at 8.8 μM. The explants were placed in the growth chamber at the same conditions described above (Section 2.3.1. Experiment 1).

As soon as shoot induction was observed (after four weeks), four to five shoots were transferred into baby food glass jars with Magenta[TM] b-cap lids and 25 mL of EM (Supplementary Table S2), and they were subcultured every six weeks. Shoots that reached 10–15 mm were separated and individually cultivated in fresh EM. The growth chamber temperature and photoperiod were the same as those previously described (Section 2.3.1. Experiment 1).

### 2.3.4. Root Induction and Acclimatization of Rooted Plants

Shoots at least 20–25 mm long from the EM were employed for root induction. Based on the results of Experiment 1, the shoots were cultivated in Ecoboxes filled with half-strength macronutrient QL basal medium supplemented with 50 μM NAA, 1.5% (*w/v*) sucrose, and 8 gL$^{-1}$ Difco Agar® (Supplementary Table S2). In addition, based on the results from Experiment 1, red light was selected for the rooting stage; the shoots were placed under dim light for eight days, followed by four weeks under a 16 h photoperiod with red light (peak wavelength 630 nm) and 60 μmol m$^{-2}$s$^{-1}$ light intensity provided by adjustable LEDs (RB4K Grow Light LEDs). After these five weeks, shoots were cultured for six weeks in Ecoboxes with 100 mL of REM (Supplementary Table S2). The photoperiod and the temperature of the growth chamber were the same as those described in previous sections. Then, rooted plants were acclimatized as described above (Section 2.3.2).

### 2.4. Data collection and Statistical Analysis
#### 2.4.1. Experiment 1

Twenty-four to forty-eight test tubes and one explant per test tube (AT, AS, BT, or BS) per each tree (two trees) were cultured in each culture medium. After two months of culture, the contamination, survival, and shoot induction percentages for each condition tested were measured. In the case of the shoot induction (%) and the mean number of shoots per explant (NS/E), these were calculated with respect to the non-contaminated explants. The effect of the explant type and culture medium on survival and shoot induction (%) was analyzed using a logistic regression model, and when necessary, differences were assessed by Tukey's post hoc test ($\alpha = 0.05$).

Data for the NS/E were analyzed by analysis of variance (ANOVA). When necessary, multiple comparisons were made using Tukey's post hoc test ($\alpha = 0.05$).

After the root expression stage, data for the root induction percentage, the mean number of roots per explant (NR/E), and the length of the longest root (LLR) (cm) were recorded. A completely randomized design using seven to twenty-four plantlets per sucrose concentration and light treatment was performed.

The effect of the sucrose concentration and light treatment on root induction (%) was analyzed with a logistic regression model. Data for NR/E and LLR were analyzed by ANOVA, and when necessary, differences were assessed by Tukey's post hoc test ($\alpha = 0.05$). To evaluate the effect of the sucrose concentration and light treatment on the acclimatization percentage, a logistic regression model was applied to plantlets after four weeks of growth in the greenhouse. Data processing was done using R Core Team software® (version 4.2.1, Vienna, Austria).

#### 2.4.2. Experiment 2

Forty test tubes and one explant per test tube (BT) were cultured in each culture medium per tree (five trees). After two months of culture, the contamination, survival, and shoot induction percentages were recorded for each condition tested.

The shoot induction percentage (%) and the mean number of shoots per explant (NS/E) were calculated with respect to the non-contaminated explants after the elongation stage. The effect of the cytokinin type on survival and shoot induction (%) was analyzed using

a logistic regression model; when necessary, differences were assessed by Tukey's post hoc test ($\alpha$ = 0.05).

Data for the NS/E were analyzed by analysis of variance (ANOVA), and when necessary, differences were assessed by Tukey's post hoc test ($\alpha$ = 0.05).

After the root expression stage, data for the root induction percentage, the mean number of roots per explant (NR/E), and the length of the longest root (LLR) (cm) were recorded. A completely randomized design using forty-one to fifty plantlets per cytokinin type applied during the shoot induction stage was performed.

The effect of the cytokinin type applied during the shoot induction stage on the root induction was analyzed using a logistic regression model. Data for NR/E and LLR were analyzed by ANOVA, and when necessary, differences were assessed by Tukey's post hoc test ($\alpha$ = 0.05). To evaluate the effect of the cytokinin type applied during the shoot induction stage on the acclimatization percentage, a logistic regression was used. As mentioned above, all data were processed using R Core Team software®.

## 3. Results

### 3.1. Micropropagation Process

#### 3.1.1. Experiment 1

The rates of contamination were registered eight weeks after sterilization, showing general values of 39%. Explants' survival was significantly affected by the explant type used; AS explants showed significantly higher survival rates (90%) than AT (62%) and BT (32%; Figure 2, Supplementary Table S3).

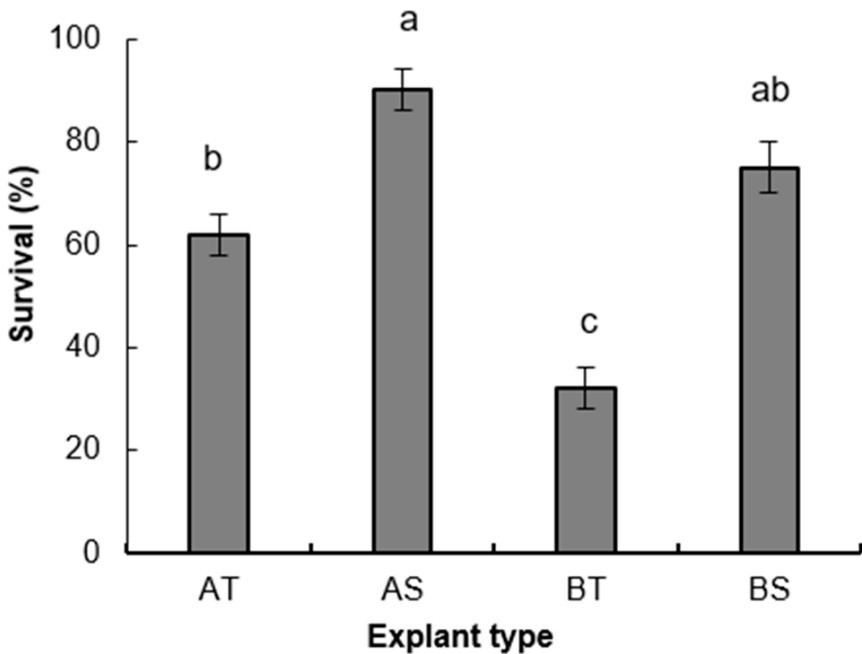

**Figure 2.** Survival (%) in different explant types of *Cryptomeria japonica* cultured in DCR medium [13], MS medium [14], and QL medium [15]. Apical explants > 1.5 cm (AT), apical explants < 1.0 cm (AS), basal explants > 1.5 cm (BT), and basal explants < 1.0 cm (BS). Data are presented as mean values ± S.E. Significant differences are indicated by different letters according to Tukey's post hoc test ($p$ < 0.05).

The basal medium and the interaction between explant type and basal medium did not show significant differences for survival (%) (Supplementary Table S3). The survival percentage ranged from 57% in explants grown in MS and QL media to 63% in explants cultured in DCR medium.

When the variables explant type and basal medium and the interaction between them were analyzed after the induction stage, statistically significant differences were only

found for the percentage of shoot induction depending on the explant type (Figure 3 and Supplementary Table S4). A significantly higher shoot induction percentage was obtained in AS (69%) and AT (56%), compared to the rest of the explants tested (Figure 3).

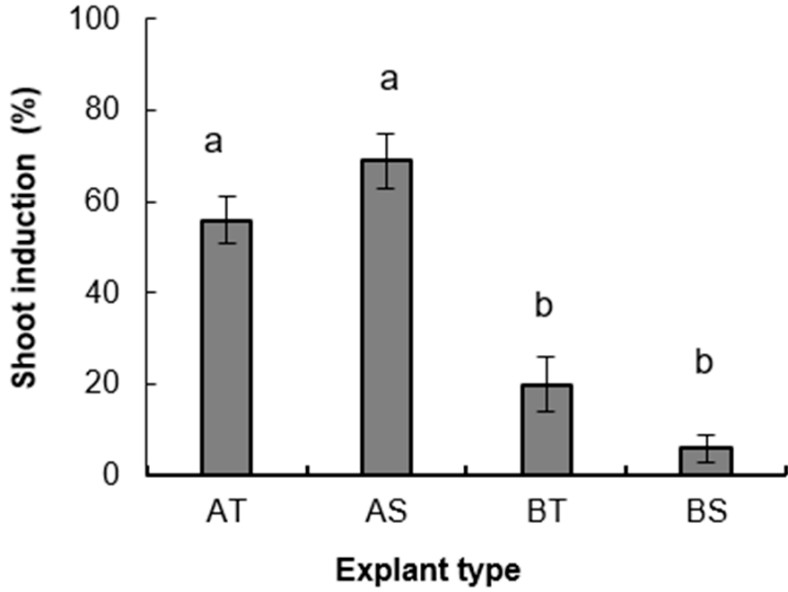

**Figure 3.** Shoot induction (%) in different explant types of *Cryptomeria japonica* cultured in DCR medium [13], MS medium [14], and QL medium [15]. Apical explants > 1.5 cm (AT), apical explants < 1.0 cm (AS), basal explants > 1.5 cm (BT), and basal explants < 1.0 cm (BS). Data are presented as mean values ± S.E. Significant differences are indicated by different letters according to Tukey's post hoc test ($p < 0.05$).

The shoot induction percentage ranged from 38% in explants grown in DCR medium to 46% in explants cultured in QL medium. Explants developed in MS medium showed an intermediate value of shoot induction (43%).

Regarding the NS/E, significant differences were found for the variables explant type and basal medium and the interaction between them (Figure 4, Supplementary Table S4). BT explants cultured in QL medium produced a significantly higher response than the other explant and medium combinations tested (Figure 4). Based on the results of Experiment 1, basal explants of >1.5 cm length and QL medium were selected to carry out the micropropagation process in Experiment 2.

3.1.2. Root Induction and Acclimatization of Rooted Plants

When the effect of the sucrose concentration, the light treatment, or the interaction between them on the root induction (%) was analyzed, no statistically significant differences were observed. (Supplementary Table S5). The root induction percentage ranged from 36% in shoots cultured in 3% sucrose concentration under FL to 54% in shoots, independently of the sucrose concentration in the culture medium and exposure to red LEDs.

The root number was significantly affected by the light treatment applied in the rooting phase (Figure 5 and Supplementary Table S5). In this sense, explants exposed to red LEDs showed significantly higher NR/E (6.5 ± 0.5) than those under fluorescent light (2.7 ± 0.4) (Figure 5). The sucrose concentration and the interaction between sucrose concentration and light treatment did not show statistically significant differences (Supplementary Table S5). The NR/E ranged from 4.5 ± 0.6 for shoots grown with 1.5% sucrose to 5.4 ± 0.7 for shoots cultured in QL medium supplemented with 3% sucrose.

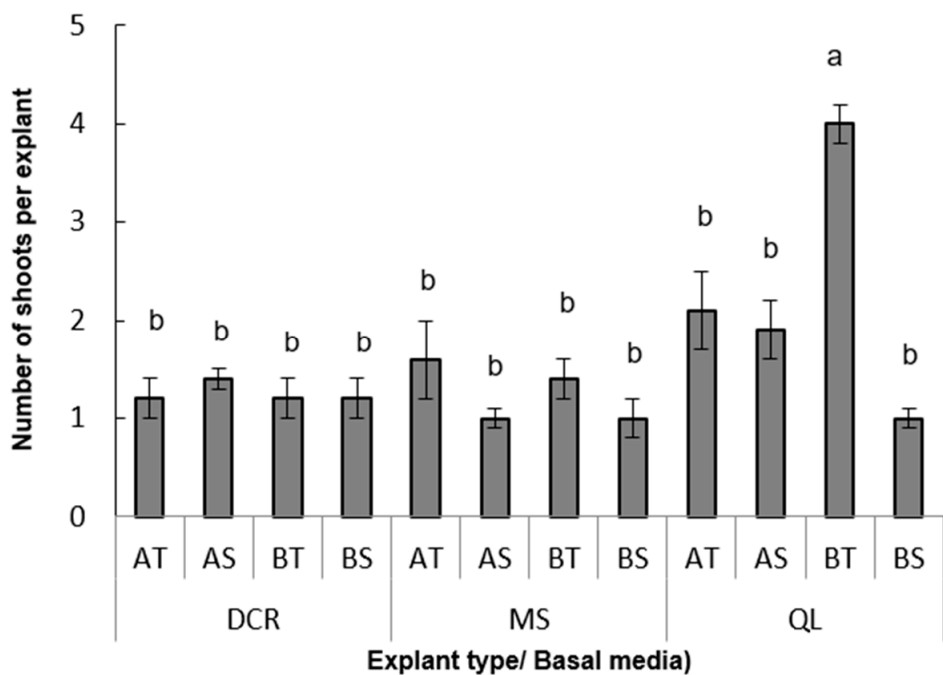

**Figure 4.** Number of shoots per explant in different explant types of *Cryptomeria japonica* cultured in DCR medium [13], MS medium [14], and QL medium [15] supplemented with 6-benzyladenine (BA, 8.8 μM). Apical explants > 1.5 cm (AT), apical explants < 1.0 cm (AS), basal explants > 1.5 cm (BT), and basal explants < 1.0 cm (BS). Data are presented as mean values ± S.E. Significant differences are indicated by different letters according to Tukey's post hoc test ($p < 0.05$).

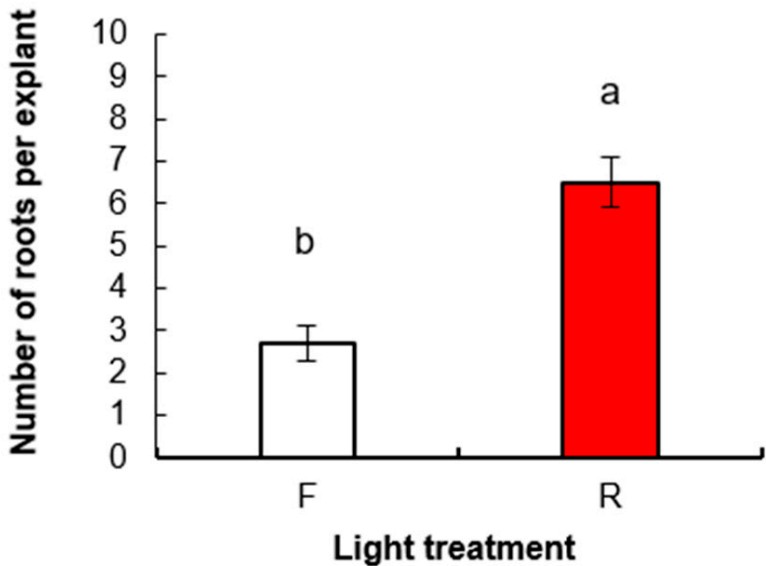

**Figure 5.** Number of roots per explant in shoots of *Cryptomeria japonica* cultured in QL medium [15], supplemented with 3% (*w/v*) sucrose or 1.5% (*w/v*) sucrose under light treatments (fluorescent light (F) and red LEDs (R)). Data are presented as mean values ± S.E. Significant differences are indicated by different letters according to Tukey's post hoc test ($p < 0.05$).

The different sucrose concentrations (1.5% or 3.0%) and light treatments tested for shoot induction showed a statistically significant effect on LLR, whereas the interaction between them did not have a significant effect (Figure 6 and Supplementary Table S5). A significantly higher LLR was observed in shoots cultured in the presence of 3% (*w/v*) sucrose compared with shoots grown at the lowest sucrose concentration (Figure 6a). In

the same way, shoots exposed to fluorescent light showed significantly higher LLR than those under red LEDs (Figure 6b).

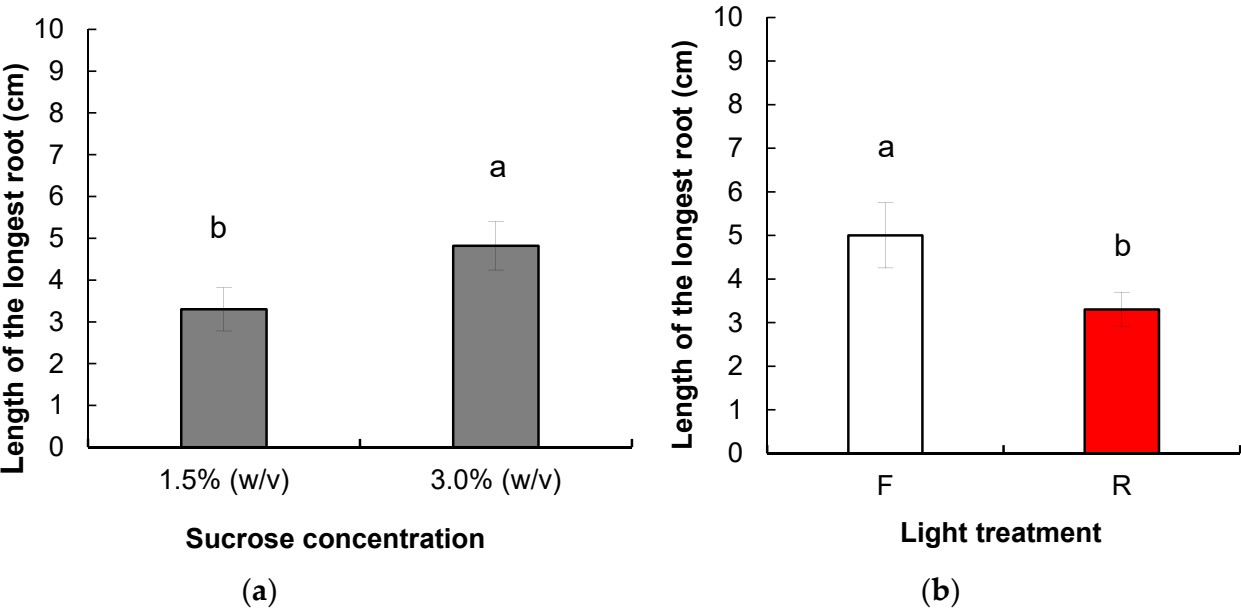

**Figure 6.** Length of the longest root in shoots of *Cryptomeria japonica* cultured in QL medium [15], according to sucrose concentration (3% and 1.5% (*w/v*) (**a**) and light treatments (fluorescent light (F) and red LEDs (R)) (**b**). Data are presented as mean values ± S.E. Significant differences are indicated by different letters according to Tukey's post hoc test ($p < 0.05$).

When the effect of sucrose concentration and light treatment applied during the rooting phase on the acclimatization percentage of rooted shoots was analyzed, no statistically significant differences were observed (Supplementary Table S6). Statistically significant differences were found in the acclimatization percentage for the interaction between sucrose concentration and light treatment (Supplementary Table S6). Nevertheless, as the p-value was bordering on significance, Tukey's post hoc test could not detect them. The acclimatization percentage ranged from 30% in shoots cultured in medium supplemented with 3% (*w/v*) and exposed to FL to 80% in those growing in the same sucrose concentration supplemented the culture medium under red LEDs.

*3.2. Experiment 2*

Four weeks after sterilization, the contamination rates were at 28%. No statistically significant differences were observed for the explant survival percentage considering the CK type tested (Supplementary Table S7). Explant survival percentages ranged from 66% in explants cultured in medium with m-T to 76% in explants grown with BA treatment.

The CK type showed a significant effect on the shoot induction (%) (Figure 7, Supplementary Table S8). A significantly higher shoot induction percentage was observed in explants induced with BA and m-T treatments (50% and 48%, respectively) than in explants grown on QL medium supplemented with TDZ.

No statistically significant differences were observed when the effect of CK type on the NS/E was analyzed (Supplementary Table S8). It was not possible to obtain shoots from explants induced with TDZ treatment due to tissue necrosis. Explants induced with m-T and BA treatments produced 2.3 ± 0.1 and 2.4 ± 0.1 NS/E, respectively.

Regarding the root induction percentage, significant differences were observed depending on the CK used for shoot induction (Figure 8 and Supplementary Table S9). A significantly higher root induction percentage was recorded in shoots induced with m-T (54%) than in shoots grown with BA (33%) during the shoot induction stage (Figure 8).

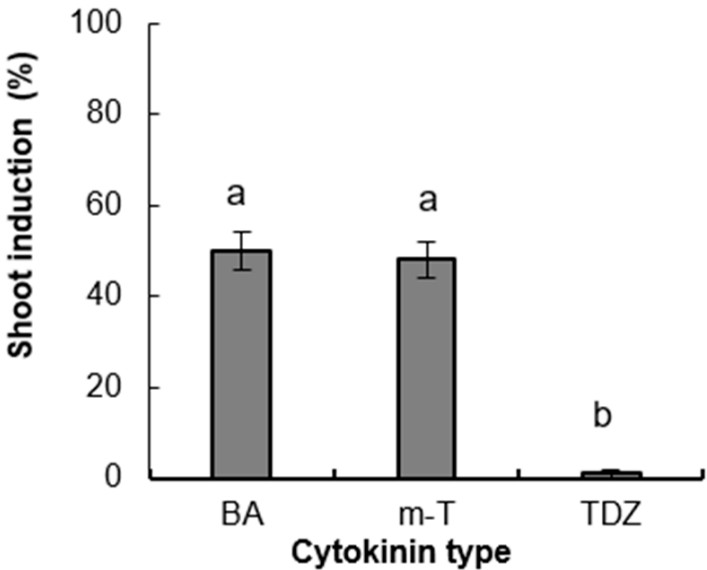

**Figure 7.** Shoot induction (%) in explants of *Cryptomeria japonica* cultured on QL medium [15], supplemented with 6-benzyladenine (BA), meta-topolin (m-T), and thidiazuron (TDZ) (8.8 µM). Data are presented as mean values ± S.E. Significant differences are indicated by different letters according to Tukey's post hoc test ($p < 0.05$).

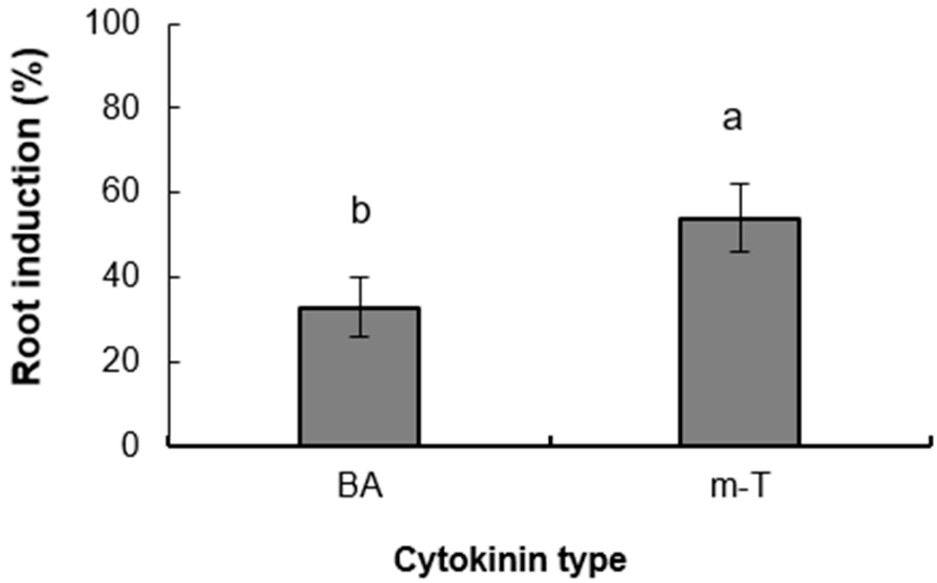

**Figure 8.** Root induction (%) in shoots of *Cryptomeria japonica* cultured in QL medium [15], supplemented with 50 µM 1-naphthaleneacetic acid (NAA), according to cytokinin type (6-benzyladenine (BA) and meta-topolin (m-T) (8.8 µM) use during shoot induction stage. Data are presented as mean values ± S.E. Significant differences are indicated by different letters according to Tukey's post hoc test ($p < 0.05$).

When the effect of the CK type utilized for shoot induction was evaluated for the NR/E and LLR parameters, no statistically significant differences were observed (Supplementary Table S9). The NR/E was $3.1 \pm 0.5$ for shoots previously induced with BA and $3.9 \pm 0.6$ for those from m-T treatment.

Explants cultured with BA and m-T treatment during the induction stage showed a LLR of $3.3$ cm $\pm 0.5$ and $3.7$ cm $\pm 0.3$, respectively.

No statistically significant differences were observed for shoots coming from different CK treatments when acclimatization percentage was analyzed (Supplementary Table S10). The ex vitro survival rate was 92% for shoots induced with BA and 94% for those induced with m-T treatment.

## 4. Discussion

As was reviewed in [28], contamination is considered a crucial obstacle that prohibits the successful establishment of an aseptic in vitro culture. In this study, contamination rates ranging from 28% to 39% were obtained using sodium hypochlorite. In this sense, our work obtained lower contamination rates than those recorded by [29] for *C. japonica*, who obtained 46% using calcium hypochlorite for surface sterilization of apical explants from adult trees. The sterilization protocol applied in Experiment 1 and Experiment 2 resulted in an optimal in vitro establishment of *C. japonica.* Therefore, our results suggest that it can be used for the establishment of cultures from other mature conifer explants [17].

Factors such as the type, size, or age of the explant, the physiological state of the donor plant, and the type of disinfectant and its concentration can influence the effectiveness of the sterilization protocol [30]. In our case, the survival percentage was significantly higher in AS explants in Experiment 1. A similar tendency was observed in *Cedar libani*, where shoot apices (<1.0 cm) from adult trees showed the best survival response [31]. Similarly, in *Taxus mairei,* the highest survival percentage was obtained in small stems (<1.0 cm) from cutting [32]. In our study, the smaller size and lower exposure of the explants' surface could have favored the culture establishment [33], as supported by [34], who mentioned that smaller explants may be easier plant material to sterilize from microorganisms.

In our experiments, the efficiency of the organogenic response was assessed by considering the shoot induction percentage and the NS/E. In Experiment 1, the results showed statistically significant differences for AS and AT explants for shoot induction percentage. In this sense, our study confirmed that the explant size is an important factor affecting axillary bud proliferation [31,34], and probably, the larger explants have more mineral nutrient reserves and endogenous hormones to support the culture [35]. Additionally, George et al. [34] explained that bigger explants from extensive parts of the shoot apex or stem segments with lateral buds could have advantages over smaller explants. Furthermore, it is probable that the morphogenetic gradient, where variations in the levels of endogenous plant growth regulators such as auxins (from the basal to the apical explant region) [34] are responsible for the different morphogenesis responses of the explants tested.

Cytokinins play several recognized roles in plant development, through the suppression of apical dominance and promotion of the development of axillary buds, the promotion of cell division, or the stimulation of plant protein synthesis [16,36]. BA is the most commonly used plant growth regulator; it is applied alone or in combination with other CK to promote in vitro shoot induction due to its effectiveness and affordability [37–39]. During Experiment 2, explants cultured with BA or m-T showed an efficient organogenic response regarding shoot induction percentage, and although not significant, a slightly higher response in BT explants induced with BA was observed. Analogous results were obtained in several organogenesis protocols from adult trees of *Pinus pinea*, *P. radiata*, *Sequoia sempervirens,* and *P. halepensis,* where, to obtain an in vitro shoot response, BA at concentrations ranging from 4.4 to 50 µM was utilized [10,40–43].

The highest NS/E was obtained in BT explants cultured in QL medium (Experiment 1). In accordance with this result, in previous experiments in our lab, the highest number of NS/E was obtained in *S. sempervirens* when explants bigger than 1.5 cm in length were used [10]. Similarly, Hine et al. [29] and Rafi and Salehi [44] developed an in vitro protocol for *C. japonica* and *Cedrus deodara,* using explants of mature trees from 1.0 to 2.5 cm in length, respectively. On the other hand, NS/E was higher on basal QL medium than on basal MS and DCR medium. According to Maruyama et al., 2021 [45], this may be attributable to a higher level of inorganic nitrogen present in those media when compared to QL medium. This hypothesis is supported by Tuskan et al. [46] who explained that a nitrate excess

could have a negative impact on the organogenic response. Similar to our experiment on *C. japonica*, the lower nitrogen content of QL medium promoted organogenesis in *Cedrus deodara* and *P. ponderosa* [44,47]. Recently, for improving the micropropagation protocol of plants, artificial neural networks algorithms have been used to build models to determine the effect of mineral nutrients, vitamins, and plant growth regulators on several growth and quality parameters of micropropagated plants [48].

Several studies attributed the improvement in multiplication rates or rooting percentages and the alleviation of physiological disorders to the use of topolins in plant tissue culture [49]. However, in our study, no significant differences for the effect of CK on NS/E were found, but a slightly higher response in explants cultured with BA was observed (Experiment 2). Analogous results were obtained in *P. radiata*, *S. sempervirens*, and *P. ponderosa*, where the induction of axillary buds was not improved when applying m-T instead of BA [10,41,47,50].

In recent years, TDZ has received more attention due to its ability to aid in vitro regeneration of woody plants [17,40,51], and it has been recommended in explants coming from adult trees to induce regeneration via axillary shoot proliferation and adventitious shoot organogenesis [52]. However, in our work, it was not possible to obtain shoots coming from explants induced with TDZ due to tissue necrosis. The reason for this result may be due to an inadequacies concentration of TDZ, as supported by [53], who mentioned that the TDZ concentration and exposure time at their optimum depends on the species. Nevertheless, TDZ at low concentrations, in pulse treatments, or during short exposure periods can be effective in circumventing TDZ-induced abnormalities such as tissue necrosis [53].

Adventitious rooting can be influenced by physical and chemical factors, among them plant growth regulators, light quality, temperature, medium composition, and carbohydrates [51]. When root induction percentage was analyzed, no significant effect of sucrose concentration (1.5 or 3.0%) and light treatment was found. A similar result was observed in the organogenesis protocols of *C. japonica*, where the culture medium was supplemented with sucrose (from 1.5 to 3.0%) to obtain rooting [5,29,54].

As mentioned above, in the last few years, m-T has been proven as an alternative to conventional CK for in vitro propagation of plants [55]. In this sense, the advantages of m-T to promote rhizogenesis have been described [49]. In Experiment 2, when we focused on the study of the effect of the cytokinin type on root induction, we found that shoots induced with m-T promoted the highest rooting percentage. Similar to our results, Naaz et al. [56], studying *Syzygium cumini*, reported that shoots induced with m-T increased rhizogenic competency compared with shoots coming from kinetin, 2-isopentyl adenine, or BA. Similarly, in *Caralluma umbellata*, shoots derived from a culture medium supplemented with m-T and NAA showed the highest in vitro rooting activity [57]. In contrast to our work, in *S. sempervirens* and *Juniperus drupacea*, growing shoots with m-T did not improve the rooting response [10,58]. Summarizing, the effect of m-T to promote rhizogenesis at different concentrations may be species-specific and depends on the starting material used to establish the in vitro culture.

Plant growth and development are strongly influenced by the quality of the light in their environment [59]. In the last few years, LEDs have shown a favorable response in in vitro culture when compared with the results obtained using fluorescent tubes [60,61]. Additionality, Ragonezi et al. [62] mentioned that the light type and wavelength specificity influence adventitious rooting. In this sense, statistically significant differences in NR/E were found when shoots were cultured under red LEDs (Experiment 1). Similarly, in *C. japonica* and *Populus sieboldii* × *Populus grandidentata*, red LEDs showed a higher response to in vitro rooting [5,63]. In *Pinus pseudostrobus* shoots exposed to red LEDs, the best rooting rates were observed at 30 days of evaluation [26]. Contrary to our results, *P. radiata* and *P. ponderosa* shoots grown under white fluorescent light and white LEDs, respectively, displayed the highest rooting responses [43,47]. Regarding LLR, shoots under fluorescent light showed longer primary roots. Analogous results were observed by Ishii et al. [64] in *C.*

*japonica* and Rojas-Vargas et al. [47] in *P. ponderosa,* where shoots growing under fluorescent light showed the longest root length. Contrary to our result, the longest roots were recorded in plants of *P. radiata* and *P. pseudostrobus* exposed to red LEDs, [26,43]. Summarizing, different plant species respond differently, even when using the same light treatment, which may be due to the genotypic characteristics of the plants or the physiological state of the explants [30,65].

Regarding the acclimatization stage, no significant differences were observed for the effect of CK on acclimatization percentage, but a slightly higher response in explants cultured with m-T was observed. These results agreed with those observed in *Aloe polyphylla*, *S. cumini,* and *C. umbellate,* where the plantlets induced with m-T were successfully acclimatized [39,56,57]. The reason for this may be due to the rapid uptake and transport of mT into the plant system and the production of reversibly sequestered metabolites [57].

## 5. Conclusions

The regeneration of *C. japonica* through micropropagation of adult trees using basal explants of >1.5 cm length was achieved and depended on physico-chemical factors. The optimal result in terms of shoot induction was obtained when basal explants were cultured in QL medium supplemented with BA treatment.

Our results suggest that the use of m-T and a 1.5% sucrose concentration favored root induction. In this sense, the use of red LEDs was better for the number of roots per explant. Finally, in the greenhouse, the shoots, independently of the cytokinins used in the shooting stage, showed high acclimatization success. In order to optimize the micropropagation efficiency in *C. japonica,* our results suggest the use of fluorescent light for the shoot induction stage and red LEDs for the rooting stage.

**Supplementary Materials:** The following supporting information can be downloaded at: https://www.mdpi.com/article/10.3390/f14040743/s1, Table S1: Variations of basal DCR [13], MS [14] and QL [15] media tested at different stages of *Cryptomeria japonica* micropropagation process, Experiment 1; Table S2: Variations of basal QL [15] medium tested at different stages of *Cryptomeria japonica* micropropagation process, Experiment 2; Table S3: Statistical analysis for the survival (%) showed in *Cryptomeria japonica* per explant type (apical explants > 1.5 cm, apical explants < 1.0 cm, basal explants > 1.5 cm, and basal explants < 1.0 cm) and basal media (DCR [13], MS [14] and QL [15]); Table S4: Statistical analysis for shoot induction (%) and number of shoots per explant showed in *Cryptomeria japonica* per explant type (apical explants > 1.5 cm, apical explants < 1.0 cm, basal explants > 1.5 cm, and basal explants < 1.0 cm) and basal media (DCR [13], MS [14] and QL [15]); Table S5: Statistical analysis for root induction (%), number roots per explant and length of the longest root of *Cryptomeria japonica* shoots cultured in QL medium [15], supplemented with 3% (*w/v*) sucrose or 1.5% (*w/v*) sucrose, according to light treatment; Table S6: Statistical analysis for the survival (%) of rooted shoots propagated in vitro coming from *Cryptomeria japonica* adult trees, after four weeks under ex vitro conditions; Table S7: Statistical analysis for the survival (%) showed in *Cryptomeria japonica* explants (basal explants of >1.5 cm length) cultured in QL medium [15], supplemented with 6-benzyladenine (BA), meta-topolin (m-T) or thidiazuron (TDZ) at 8.8 μM; Table S8: Statistical analysis for shoot induction (%) and number of shoots per explant showed in *Cryptomeria japonica* explants (basal explants of >1.5 cm length) cultured in QL medium [15], supplemented with 6-benzyladenine (BA), meta-topolin (m-T) or thidiazuron (TDZ) at 8.8 μM; Table S9: Statistical analysis for root induction (%), number root per explant and length of longest root showed in *Cryptomeria japonica* explants (basal explants of >1.5 cm length) cultured in QL medium [15], supplemented with 50 μM 1-naphthaleneacetic acid (NAA), according to cytokinin type [6-benzyladenine (BA), meta-topolin (m-T) or thidiazuron (TDZ) at 8.8 μM]; Table S10: Statistical analysis for the survival (%) in *Cryptomeria japonica* explants (basal explants > 1.5 cm) cultured in QL medium [15], supplemented with 50 μM 1-naphthaleneacetic acid (NAA), according to cytokinin type [6-benzyladenine (BA) or meta-topolin (m-T) at 8.8 μM].

**Author Contributions:** Conceptualization, P.M. and I.A.M.; methodology, P.M. and I.A.M.; formal analysis, A.R.-V.; investigation, A.R.-V.; data curation, A.R.-V.; writing—original draft preparation, A.R.-V.; writing—review and editing, A.R.-V., P.M. and I.A.M. All authors have read and agreed to the published version of the manuscript.

**Funding:** This research has been funded by Universidad Nacional de Costa Rica and Instituto de Investigación y Servicios Forestales (INISEFOR), DECO (Basque government), MINECO (AGL2016-76143-C4-3R), MICINN (PID2020-112627RB-C32), CYTED (P117RT0522), and MULTIFOREVER project, under the umbrella of ERA-NET Cofund ForestValue by ANR (FR), FNR (DE), MINCyT (AR), MINECO-AEI (ES), MMM (FI), VINNOVA (SE), ForestValue has received funding from the European Union's Horizon 2020 Research and Innovation Programme under grant agreement no. 773324.

**Acknowledgments:** The authors sincerely thank Ander Isasmendi (NEIKER-BRTA) for his technical assistance.

**Conflicts of Interest:** The authors declare no conflict of interest.

**Abbreviations**

ANOVA—analysis of variance; AS—apical explants < 1.0 cm; AT—apical explants > 1.5 cm; BA—6-benzyladenine; BS—basal explants < 1.0 cm; BT—basal explants > 1.5 cm; CK—cytokinin; EM—elongation medium; FL—fluorescent light; IM—induction medium; LEDs—light-emitting diodes; LLR—length of the longest root; DCR—DCR medium (Gupta and Durzan, 1985); m-T—meta-topolin; MS—MS medium (Murashige and Skoog, 1962); NAA—1-naphthalene acetic acid; NR/E—number of roots per explant; NS/E—number of shoots per explant; QL—QL medium (Quoirin and Lepoivre, 1977); REM—root expression medium; RIM—root induction medium; TDZ—thidiazuron.

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
