# Peer review of "Adult Trees Cryptomeria japonica (Thunb. ex L.f.) D. Don Micropropagation: Factors Involved in the Success of the Process"

_forests, doi:10.3390/f14040743_

Round 1

Reviewer 1 Report

forests-2295656 - Review report

Adult trees Cryptomeria japonica micropropagation: factors involved in the success of the process.

Comments and Suggestions for Authors

I have carefully read the manuscript FORESTS-2295656. The ms describes a protocol for C. japonica to optimize the micropropagation method using trees of more than 30 years as source of plant material.

In my opinion, the experimental work was correctly structured but some revisions and modifications in the text are strongly needed.

In the introduction, please try to engage the reader with your work a little more: consider highlighting the problem and novelty a bit more.

In the section “Material and Methods”, several treatments combining culture media, plant growth regulators, sucrose concentrations and light treatments were tested. The reader may get a little lost among all the various treatments and their abbreviations . Therefore, it would be very helpful if the suggested protocols could be added as a summary with one or more tables.

In Figure 1b (line 93), please specify the type of explants (see lines 106-108).

References are correctly reported in the article body as well as in the dedicated section.

Author Response

Dear Editor,

We would like to thank the reviewers for their comments and suggestions as we believe they have contributed greatly to improve the manuscript.

Below you can find the response to the comments made by the reviewer 1

Best regards.

Response to Reviewer 1 Comments

GENERAL CONCEPT COMMENTS / MINOR CHANGES:

Point 1: In the introduction, please try to engage the reader with your work a little more: consider highlighting the problem and novelty a bit more.

Response 1: The comment made by the reviewer has been incorporated in introduction section.

Point 2: In the section “Material and Methods”, several treatments combining culture media, plant growth regulators, sucrose concentrations and light treatments were tested. The reader may get a little lost among all the various treatments and their abbreviations. Therefore, it would be very helpful if the suggested protocols could be added as a summary with one or more tables.

Response 2: The comment made by the reviewer has been incorporated in supplementary materials (Table 1 and Table 2).

Point 3: In Figure 1b (line 93), please specify the type of explants (see lines 106-108).

Response 2: The comment made by the reviewer has been incorporated in the Figure 1b (line 99).

Reviewer 2 Report

A manuscript titled Adult trees Cryptomeria japonica micropropagation: factors in-2 volved in the success of the process seems like good paper. In general, this manuscript has a valuable topic. The topic is scientifically sound. Experimental design is adequate. However, the manuscript needs to be corrected. Please find here some questions and suggestions that may be useful in improving the submitted manuscript.

1.       What is the novelty of this paper as compared to many other reported papers in this field?

2.       Can the author describe in detail the course of the in vitro culture. What parameters did the shoots have to achieve to transfer them to fresh media.

3.       The author could explain the difference between the three media used (DCR, MS, QL).

4.       It would be good to attach a photo of the plants as evidence showing the differences in the morphological parameters of the aboveground parts with different treatments.

5.       I did not find information on the results from the three compared media (Fig. 2 or Fig. 3)

6.       Why did the author choose such concentrations of CK used? Why wasn't a higher concentration of TDZ used despite the changes that were observed?

7.       Can the author explain why, despite such a low shoot induction, the BT explants turned out to be the best in terms of NS/E? And why AS explants did not show good results in this regard?

8.       Keywords could be revised. They do not quite represent the manuscript.

9.       The author uses many reagents in the experiments but did not include information about the companies. Author should complete this.

10.   Authors should check that all References items are describe in accordance with the requirements of the Editorial Board. References should be described depending on the type of work.

Author Response

Dear Editor,

We would like to thank the reviewers for their comments and suggestions as we believe they have contributed greatly to improve the manuscript.

Below you can find the response to the comments made by the reviewer 2.

Best regards.

Response to Reviewer 2 Comments

GENERAL CONCEPT COMMENTS / MINOR CHANGES:

 Point 1: What is the novelty of this paper as compared to many other reported papers in this field?

Response 1: The analysis of the effect of combined morphological (type of explants), chemical (culture media, sucrose concentrations) and physical factors (LED lights) in different micropropagation stages of adult trees has not been carried out in Cryptomeria japonica. Moreover, the use of ventilated culture vessels is used in our laboratory but it is not widely use in conifers species. As reported in our study, the modifications made in the protocol for this species have improved the published results using tress of more than 20 years.

 Point 2: Can the author describe in detail the course of the in vitro culture. What parameters did the shoots have to achieve to transfer them to fresh media.

Response 2: Our protocol consists firstly of all in an in vitro introduction of the material. Once the culture is established, we begin the induction of the growth of the axillary buds and their subsequent elongation. Finally, the elongated shoots are subjected to a rooting phase to be later transferred to the greenhouse to undergo their acclimatization.

Shoots were transferred to the fresh media (elongation stage) when the bud had emerged from the primordia (after four weeks in induction medium). Then, when shoots had elongated more than 20 mm long, they were transferred to root induction media.

 Point 3: The author could explain the difference between the three media used (DCR, MS, QL).

Response 3: All the media used in this study are sufficiently described in several articles. However, in the discussion section, we mentioned the differences in the level of inorganic nitrogen in all culture media described and its possible effect on the organogenesis process.

Point 4: It would be good to attach a photo of the plants as evidence showing the differences in the morphological parameters of the aboveground parts with different treatments.

Response 4: We did not observe differences in the morphological parameters of the aerial parts of the plantlets. For this reason, we consider that the introduction of another picture is not necessary.

Point 5: I did not find information of the results from the three compared media (Fig. 2 or Fig. 3)

Response 5: We found that the basal medium did not show significant differences for the survival and shoot induction (%), this information can be found in supplementary tables 3 and 4. For the abovementioned reason, the results of survival and shoot induction (%) per basal media were described only in text.

 Point 6: Why did the author choose such concentrations of CK used? Why wasn't a higher concentration of TDZ used despite the changes that were observed?

Response 6: Several studies have been developed to obtain organogenesis protocols in conifers species in our lab. In this sense, we have achieved procedures to propagate different conifers (Pinus pinea, P. radiata, Sequoia sempervirens, Pinus pinaster, Pinus sylvestris and P. halepensis) using cytokinin concentrations ranging from 2.5 to 50 µM. Based on our experience, we decided to test these concentrations.  In future experiments, it would be interesting the use of higher concentration of TDZ to analyse its effect and increase the success of the process.

Point 7: Can the author explain why, despite such a low shoot induction, the BT explants turned out to be the best in terms of NS/E? And why AS explants did not show good results in this regard?

Response 7: In the discussion section, we mentioned that probably the explant size affected axillary bud proliferation, because larger explants have more mineral nutrient reserves and endogenous hormones to support the culture. Furthermore, it is probable that the morphogenetic gradient of phytohormones, where variations in the levels of endogenous plant growth regulators such as auxins (from the basal to the apical explant region), are responsible for the different morphogenesis responses of explants tested.

Point 8: Keywords could be revised. They do not quite represent the manuscript.

Response 8: Keywords section has been modified.

Point 9: The author uses many reagents in the experiments but did not include information about the companies. Author should complete this.

Response 9: Information about the companies of different reagents used in the experiments has been incorporated in Materials and Methods section.

Point 10: Authors should check that all References items are describe in accordance with the requirements of the Editorial Board. References should be described depending on the type of work.

Point 10: References section have been modified according with the requirements of the Journal.